# HIGD-Driven Regulation of Cytochrome *c* Oxidase Biogenesis and Function

**DOI:** 10.3390/cells9122620

**Published:** 2020-12-06

**Authors:** Alba Timón-Gómez, Emma L. Bartley-Dier, Flavia Fontanesi, Antoni Barrientos

**Affiliations:** 1Department of Neurology, University of Miami Miller School of Medicine, Miami, FL 33136, USA; axt809@med.miami.edu; 2Department of Biochemistry and Molecular Biology, University of Miami Miller School of Medicine, Miami, FL 33136, USA; elb131@miami.edu (E.L.B.-D.); ffontanesi@med.miami.edu (F.F.)

**Keywords:** HIGD1A, HIGD2A, Rcf1, Rcf2, Hypoxia Inducible Gene Domain, cytochrome *c* oxidase, mitochondrial respiratory chain complex IV

## Abstract

The biogenesis and function of eukaryotic cytochrome *c* oxidase or mitochondrial respiratory chain complex IV (CIV) undergo several levels of regulation to adapt to changing environmental conditions. Adaptation to hypoxia and oxidative stress involves CIV subunit isoform switch, changes in phosphorylation status, and modulation of CIV assembly and enzymatic activity by interacting factors. The latter include the Hypoxia Inducible Gene Domain (HIGD) family yeast respiratory supercomplex factors 1 and 2 (Rcf1 and Rcf2) and two mammalian homologs of Rcf1, the proteins HIGD1A and HIGD2A. Whereas Rcf1 and Rcf2 are expressed constitutively, expression of HIGD1A and HIGD2A is induced under stress conditions, such as hypoxia and/or low glucose levels. In both systems, the HIGD proteins localize in the mitochondrial inner membrane and play a role in the biogenesis of CIV as a free unit or as part as respiratory supercomplexes. Notably, they remain bound to assembled CIV and, by modulating its activity, regulate cellular respiration. Here, we will describe the current knowledge regarding the specific and overlapping roles of the several HIGD proteins in physiological and stress conditions.

## 1. Introduction

Mitochondrial cytochrome *c* oxidase (COX) or respiratory chain complex IV (CIV) is the primary site of cellular oxygen consumption and essential for aerobic energy generation in the chemical form of ATP [1]. Mitochondrial cytochrome *c* oxidase is a copper-heme A oxidase formed by three conserved catalytic core membrane subunits (COX1, COX2, and COX3) encoded in the mitochondrial DNA and a variable number of supernumerary subunits, 8 in yeast or 11 in mammals, encoded in the nuclear genome [2]. The core subunits COX1 and COX2 contain the copper and heme A redox metal active centers of the enzyme and drive catalysis. Whereas the catalytic core is conserved in the bacterial enzyme, the addition of supernumerary subunits occurred sequentially during evolution as an adaptation to cope with evolving oxygen-rich atmosphere and eukaryotic cell energetic requirements and to regulate CIV activity and cellular respiration accordingly [3]. Some of these CIV subunits exist in several isoforms that are tissue-specific in mammals or preferentially expressed under stress [4,5,6,7].

Assembly of functional CIV involves the coming together of proteins of dual genetic origin, synthesized in two different translational machineries. Their concerted accumulation within mitochondria is regulated at the transcriptional and post-transcriptional levels and requires nucleus-mitochondria crosstalk [8,9,10]. The actual assembly process involves more than forty nucleus-encoded assembly factors acting to coordinate the folding and membrane insertion of CIV subunits, assist the incorporation of metal cofactor components into the core subunits, and direct subunit assembly into a functional enzyme [2,8,11]. CIV assembly is mostly a modular process, with the biogenesis of each of the three catalytic core subunits proceeding by a relatively independent process with the assistance of subunit-specific chaperones [2,12]. It is believed that this modular assembly is particularly fundamental for subunits COX1 and COX2 to ensure the timely incorporation of their copper and heme prosthetic groups [2,13,14]. During maturation or after the three core subunits are matured, they assemble with a subset of nucleus-encoded subunits forming modules that then incorporate sequentially to yield fully assembled CIV. Recent data have shown that in mammals, two CIV assembly factors, the hypoglycemia/hypoxia inducible gene domain (HIGD) family proteins HIGD1A and HIGD2A, homologs of the yeast respiratory supercomplex factor 1 (Rcf1), form modules with nucleus-encoded COX subunits to coordinate their assembly with the core subunits [15,16,17]. Yeast Rcf1 and mammalian HIGD2A associate with COX3 to coordinate its modular assembly [15,18,19,20,21]. In addition to assembling as a standing-alone complex, CIV also forms dimers and supercomplexes (SCs) with complex III_2_ and complex I. It is becoming evident that CIV can incorporate into SCs not only as a preassembled complex, but can also undergo modular assembly in the context of SCs, a process in which the mammalian HIGD proteins and the SC assembly factor COX7A2L seem to play an important role [15,16,22].

CIV structure meets function on an extramembrane domain of COX2 that protrudes into the mitochondrial intermembrane space (IMS) to bind soluble cytochrome *c*. At the bottom of this COX2 domain, a Cu_A_ di-copper center accepts electrons from cytochrome *c* and transfers them to heme *a* in COX1, from where electrons flow to the COX1 heme *a*_3_-Cu_B_ binuclear center where O_2_ is reduced to water [2]. In summary, CIV reduces dioxygen to water with four electrons from cytochrome *c* and four protons taken up from the mitochondrial matrix, two of which are transferred across the inner membrane to contribute to the generation of the proton gradient that is used by the F_1_F_o_ ATP synthase to drive ATP synthesis, a process known as oxidative phosphorylation (OXPHOS).

CIV plays a key role in the overall regulation of mitochondrial respiration and OXPHOS to adjust ATP production to the cellular energy requirements, depending on nutritional or environmental stimuli. As a mid/long-term regulation, CIV and respiratory activity can be regulated biogenetically by controlling the abundance of enzyme units present in the mitochondrial inner membrane [23], and/or controlling the subunit isoform composition to confer different catalytic properties to the enzyme [4,5,6,7]. As a short-term regulation, CIV activity is physiologically controlled allosterically by ATP through binding to subunit COX4-1 of the mammalian enzyme (subunit Cox5a in yeast) [24], which makes CIV the respiration rate-limiting step [25]. This respiratory control mechanism is activated by cAMP-dependent phosphorylation and reversibly switched off by calcium-induced dephosphorylation of COX4-1 [26]. The molecular form of the enzyme, as a monomer or a dimer, has also been proposed to be a mechanism of control, with the CIV monomeric form possessing a higher enzymatic activity than the dimeric one [27]. Thus, the equilibrium of monomeric or dimeric CIV via reversible phosphorylation can modulate respiration and ultimately ATP synthesis [28]. Furthermore, over the last five years, evidence has been reported indicating that yeast and mammalian HIGD proteins remain bound to CIV after promoting its assembly and play regulatory roles in its catalytic activity. As such, expression of both mammalian HIGD1A and HIGD2A is induced under hypoxia [29], and mammalian HIGD1A regulates oxygen consumption, production of reactive oxygen species (ROS), and AMPK (5′ adenosine monophosphate-activated protein kinase) activity during glucose deprivation in cancer cells [30]. HIGD1A interacts with cytochrome *c* [31] and is a positive modulator of CIV [32]. Yeast Rcf1 and Rcf2 also interact with mature CIV [33,34]. At least Rcf1 regulates *Saccharomyces cerevisiae* cellular respiration by altering the COX catalytic site [35] to modulate its activity [36]. It has also been proposed that Rcf1 maintains a population of functionally competent, proton pumping CIV [37], and mediates the formation of an electron-transfer bridge from CIII to CIV via a tightly bound cytochrome *c*, which would promote efficient electron transfer in the context of SCs [38].

The HIGD (or HIG1) protein family members are characterized by the presence of an N-terminal HIG1 domain (PS51503), predicted to be integral membrane proteins consisting of two hydrophobic helices that might tend to form a hairpin-like loop across the lipid bilayer [39,40]. They can be divided into two subgroups known as HIGD type 1 and type 2 isoforms, based on whether they contain a conserved QRRQ motif (Q/I)X_3_(R/H)XRX_3_Q) (type 2), or a modified version of this motif, missing the first QR (type 1). Whereas type 2 isoforms are found in all eukaryotes and some α-proteobacteria, type 1 members are restricted to higher eukaryotes (Figure 1a,b). *S. cerevisiae* Rcf1 and Rcf2, as well as mammalian HIGD2A, are type 2 proteins, and mammalian HIGD1A is a type 1 isoform. In general, type 2 proteins such as Rcf1/2 are constitutively expressed, although HIGD2A responds to changes in oxygen concentration, glucose availability, and cell cycle [29]. On the contrary, type 1 proteins, such as HIGD1A, are generally hypoxia- and stress-induced isoforms [30,39,41].

In this manuscript, we will briefly summarize our current understanding of the mechanisms by which HIGD proteins control CIV biogenesis and will then detail how these proteins regulate CIV activity and cellular respiration in yeast and mammals in physiological, stress, and disease conditions.

## 2. Role of HIGD Proteins in Yeast Cytochrome *c* Oxidase Biogenesis and Function

### 2.1. Discovery of Rcf Proteins in Saccharomyces cerevisiae

The first link of HIGD isoforms to CIV was made by the discovery of Rcf proteins in *S. cerevisiae*. In 2009, the open reading frames Yml030w and Ynr018w were identified in a genome-wide screen for mtDNA maintenance genes and renamed Altered Inheritance of Mitochondria genes AIM31 and AIM38 [48]. The encoded proteins were shown to co-migrate with CIV in a large-scale proteomic analysis of mitochondrial extracts resolved by Blue Native (BN)-PAGE [49]. However, their function remained uncharacterized until 2012, when the work of three independent research groups provided evidence supporting a role in the biogenesis and function of the mitochondrial respiratory chain [19,20,50]. Driven by the growing interest in mitochondrial respiratory SCs, both Rosemary Stuart’s group at Marquette University and Peter Rehling group at the University of Gottingen identified Aim31 and Aim38 by mass-spectrometry analysis of yeast isolated SCs [19,20]. Following a broader approach, Jared Rutter’s group at the University of Utah aimed to characterize evolutionary conserved mitochondrial proteins of unknown function, including Aim31 [50]. The critical observation, reported by the three studies, was that the steady-state level of the respiratory SC III_2_–IV_2_ was decreased in the absence of Aim31 alone or in combination with Aim38, led to the current nomenclature of Respiratory superComplex Factors 1 and 2 (Rcf1 and 2) [19,20,50].

Rcf1 and Rcf2, HIGD family type 2 subgroup proteins, are intrinsic mitochondrial inner membrane proteins of 18 and 25 kDa, respectively [19,20]. They localize exclusively to mitochondria [50], although they lack a standard N-terminal mitochondrial targeting sequence. Whereas the HIG1 homology domain in Rcf1 is located in its N-terminal region (residues 5–96), it is found in the central region of Rcf2 (residues 89–180) (Figure 1c). In addition, fungal Rcf1 proteins contain a non-conserved C-terminal extension [50], whose function is discussed below.

Whereas both Rcf1 and Rcf2 interact with respiratory SCs, only Rcf1 specifically associates with CIV in the absence of CIII, or upon detergent-mediated dissociation of wild type SCs [19,20,50]. Rcf1 was proposed to play a dual function in the biogenesis of the mitochondrial respiratory chain, assisting CIV assembly and SC formation, as explained in the next section. The separation of these two potentially interconnected Rcf1 roles has proven to be a significant challenge in the field. Furthermore, the early observation that Rcf1 and Rcf2 did not co-purify with each other [19] led to postulate the existence of heterogeneous, Rcf1-, or Rcf2-associated, SC subpopulations. Rcf1 has also been reported to determine CIV variability based on the presence or absence of the Cox13 subunit [20]. In summary, the three initial Rcf studies brought to the central stage two fundamental concepts, which have remained the object of extensive investigation to today: the ideas of heterogeneity in the mitochondrial respiratory chain (MRC) complex and SC composition and regulation of the bioenergetics output mediated by the diversity in the MRC configuration in response to environmental changes. Indeed, MRC alterations described in the Δ*rcf1* mutant strain are associated with a modest decrease in respiratory capacity, which is largely exacerbated by the concomitant deletion of *rcf2* [19], growth under low oxygen tension [20], oxidative stress, or at elevated temperature [50].

### 2.2. Role of Rcf Proteins in Cytochrome c Oxidase Assembly and Function

The absence of Rcf1 results in decreased SC III_2_–IV_2_ levels [9,20,42], a parallel accumulation of free CIII, and a decrease in CIV levels to approximately 50% of wild type [33], suggesting a primary CIV assembly defect. Rcf1 interacts with newly-synthesized Cox3, and this interaction is preserved in an array of COX assembly mutants in which post-translational maturation of Cox1 or Cox2 is affected [19], suggesting that Rcf1 interacts with Cox3 prior to its incorporation into the holoenzyme. Rcf1 can be found as part of the Cox3 module (Figure 2a) that additionally contains at least the three nucleus-encoded COX subunits, Cox4, Cox7, and Cox13 [21]. Rcf1 is not essential for assembling the Cox3 module or its progression into fully assembled CIV [19,21]; however, it is required to optimize the efficiency of the process [21]. On the other hand, Rcf1 also enhances the incorporation of late-assembly CIV subunits Cox12 (human COX6b) and Cox13 (human COX6a) into monomeric or supercomplexed CIV [19,20], and the incorporation of Cox12/Cox13-containing CIV monomers into SCs [20,51]. Thus, Rcf1 may modulate the composition of CIV present in SCs to fine-tune energy production in response to metabolic or environmental cues. Notably, Cox12 and the CIV assembly factor Coa6, required for the maturation of the catalytic subunit Cox2, undergo genetic and physical interactions [52], suggesting that Cox12 may be part of the Cox2 assembly module and supporting the untested hypothesis that Rcf1 may play a role in the association between the Cox2 and Cox3 modules during the CIV assembly process.

Despite the association of Rcf1 with Cox3 from the early steps in CIV biogenesis to fully-assembled SCs, which is not a standard property for assembly factors, several lines of evidence argue against Rcf1 being a stoichiometric CIV structural component. Among them, Rcf1 is less abundant than CIV subunits in mitochondria, only a small fraction of Rcf1 interacts with CIV and SCs, and is specifically found in association with only a CIV subpopulation [36,51]. Instead, Rcf1 may act as a Cox3 chaperone required to promote its proper conformation, which in turn could affect the incorporation of the late assembly subunits Cox12 and Cox13 and/or the properties of the enzyme catalytic center [33,51]. Indeed, although devoid of prosthetic groups, Cox3 plays a critical role in CIV enzymatic activity. Together with a set of tightly associated phosphatidylglycerol and cardiolipin lipid moieties, Cox3 forms part of the oxygen translocation channel and can affect CIV oxygen uptake [53,54]. Additionally, through long-range interactions with Cox1, also involving lipid binding, Cox3 modulates the efficiency of proton pumping [55] and determines structural changes in the CIV active site, which stabilize the heme *a*_3_-Cu_B_ redox center and prevent CIV suicide inactivation [53]. It has been proposed that Rcf1 could mediate the correct maturation of Cox3 by facilitating the proper Cox3-associated lipid arrangement [51]. This hypothesis is supported by the lack of an additive SC instability phenotype upon *rcf1* deletion in cells lacking the cardiolipin synthase enzyme Crd1, also implicated in SC stability, suggesting that the two factors may act in the same pathway [50]. Moreover, a mutation in the conserved QRRQ motif in Rcf1 (R67A) led to an enhanced physical association between Rcf1 and Cox3 accompanied by increased sensitivity to dicyclohexylcarbodiimide (DCCD) [51]. DCCD binds to Cox3 and induces conformational changes causing the inhibition of CIV proton-pumping activity [56]. Recent bioenergetics analyses in isolated mitochondria support an Rcf1-mediated effect on proton pumping efficiency. Upon normalization by CIV amounts, both maximal and coupled NADH-driven respiration were found decreased in the absence of Rcf1, whereas leak respiration was not [33]. Consequently, the respiratory control ratio (RCR), a measurement of mitochondrial coupling calculated as coupled over leak respiration, was reduced in Δ*rcf1* mitochondria, indicating a higher degree of uncoupling between electron transfer and ATP synthesis [33]. The CIV defect in Rcf1-deficient mitochondria impairs the generation of a full proton gradient across the inner membrane [33], as suggested by two observations. First, the mitochondrial membrane potential established by respiration driven by either NADH or the cytochrome *c* synthetic electron donor ascorbate-TMPD was lower in the Δ*rcf1* mutant than in the wild type strain [33]; and second, the Δ*rcf1* strain has an increased sensitivity to the uncoupler CCCP [50].

Rcf1 also mediates structural changes in the CIV heme *a*_3_-Cu_B_ catalytic site, as inferred from the results of spectrophotometric measurements of carbon monoxide (CO) ligand binding kinetics to heme *a*_3_ in both isolated mitochondria and purified CIV. From these studies, two CIV populations were identified, one with slow kinetics of CO dissociation from CIV and another with rapid kinetics, which could be further separated into two sub-groups based on the magnitude of the perturbations observed in the absorption spectra [35,38]. Both species were present in wild type and Δ*rcf1* mitochondria. However, while in the wild type, the rapid population represents only ~10% of the total, it constitutes the predominant component in the absence of Rcf1 [35]. Moreover, the rapid CIV population was not an indirect consequence of SC destabilization since it did not increase in a strain lacking CIII [38]. These observations have suggested that, in the absence of Rcf1, only a fraction of CIV is fully active due to Rcf1-dependent structural alterations in the enzyme catalytic site. Additionally, the presence of different CIV populations in wild type mitochondria opens the possibility that through its binding to assembled holo-CIV, Rcf1 may control the proportion of fully active enzyme.

### 2.3. Role of Rcf Proteins in Yeast Respiratory Supercomplex Biogenesis and Function

Given the role of Rcf1 in CIV assembly, the lower levels of SCs in the absence of Rcf1 could simply be a consequence of the attenuated levels of CIV available to form SCs. However, in wild type cells, a small fraction of Rcf1 is found in association with SCs [19,20,50] (Figure 2b). The role of this Rcf1 pool could still be related to CIV biogenesis to facilitate the incorporation of the late-assembly CIV subunits Cox12 and Cox13 directly in the context of SCs [20] and/or by ensuring the maintenance of a proper Cox3-associated lipid complement [51]. Additionally, Rcf1 interacts independently with CIII and CIV, and it has been proposed to play a role in SC formation and/or stability [19,20,50]. Although Rcf1 usually binds to Cox3, the R67A QRRQ mutant variant also directly interacts with Cox2 [51], and, in the absence of CIV and fully assembled CIII, Rcf1 was found to interact with the CIII catalytic subunit cytochrome *c_1_* (Cyt*c_1_*) [50].

Remarkably, the CIV subunits Cox12 and Cox13, whose incorporation into SCs is modulated by Rcf1, as well as the CIV core subunits Cox2 and Cox3 and the CIII subunit Cyt*c_1_*, which physically interact with Rcf1, all participate in the formation of the binding sites for the mobile electron carrier cytochrome *c* in CIV and CIII, respectively. These interactions have significant implications for the efficiency of electron transfer from CIII to CIV. Although the Δ*rcf1* mutant displays wild type CIII and decreased CIV activities, when the coupled CIII–CIV activity was measured in the presence of ubiquinol and yeast cytochrome *c*, the kinetics of cytochrome *c* re-oxidation was faster in the absence of Rcf1 [38]. This observation was interpreted to indicate the existence of a tightly bound cytochrome *c* molecule mediating electron transfer between CIII and CIV in the presence of Rcf1 in wild type mitochondria [38]. This direct electron transfer would be faster than the equilibration of electrons with the larger cytochrome *c* pool present in the solution. On the contrary, in the absence of Rcf1, the direct electron transfer does not occur, and the cytochrome *c* molecules reduced by CIII equilibrate with the cytochrome *c* pool prior to binding to and oxidation by CIV [38]. However, the interaction between Rcf1 and cytochrome *c* has been shown only in vitro [58], and the observed behavior could be an indirect consequence of Rcf1-dependent SC instability rather than a direct Rcf1-mediated electron transfer bridge. Indeed, we have recently reported that a lack of SCs decreases cytochrome *c*-mediated electron transfer efficiency between CIII and CIV without compromising CIV maximal enzymatic activity [57].

The functions of mammalian cytochrome *c* in both electron transport and apoptosis are regulated by tissue-specific phosphorylation at defined residues [59]. The enzymatic activity of yeast CIV measured in isolated mitochondria is higher in the presence of purified human cytochrome *c* variants carrying phosphomimetic substitutions in Y48 or Y97 than wild type cytochrome *c*. In the case of the Y48 but not the Y97 phosphomimetic, this effect was not observed in Δ*rcf1* mitochondria, suggesting that Rcf1 could play a role in modulating cytochrome *c* function depending on its phosphorylation state [31,60]. However, these data should be interpreted with caution since it has been reported that yeast coupled CIII-CIV activity largely differs when exogenous yeast versus bovine cytochrome *c* is used in the assay [38].

Different from Rcf1, Rcf2 is not required for CIV assembly, and its absence has little effect on CIV content [19,20,33]. Notably, the endogenous cellular respiratory rate of the Δ*rcf2* mutant strain was significantly higher than the wild type, whereas the RCR and membrane potential measured in isolated mitochondria were decreased in the absence of Rcf2 [33]. A role in the generation of the mitochondrial membrane potential is also supported by the growth hypersensitivity of the Δ*rcf2* mutant strain to the K^+^/H^+^ ionophore nigericin that acts as a mitochondrial inner membrane-specific uncoupler [33]. Moreover, while the absence of Rcf2 does not further exacerbate the decrease in CIV levels observed in the Δrcf1 mutant strain, the lack of both HIGD proteins resulted in an additive effect on mitochondrial membrane potential attenuation [33], helping to explain the severity of the respiratory growth defect observed for the double mutant Δ*rcf1*Δ*rcf2* [19,20]. Rcf2 has an N-terminus homolog lacking the HIG1 domain, which was named Rcf3 (YBR255c-A) [61]. Rcf3 interacts with free CIV and respiratory SCs, and, as in the Δ*rcf2* strain, the absence of Rcf3 alone increases oxygen flux via CIV without affecting respiratory growth [61]. On the contrary, the double deletion Δ*rcf2*Δ*rcf3* decreases the respiratory rate and growth, thus suggesting an overlapping role of the two proteins in mitochondrial respiration [61].

The C-terminal portion of Rcf2, comprising two transmembrane α-helixes and a final C-terminal α-helix extending in the intermembrane space (IMS), has been identified in hypoxic respiratory SCs reconstructed by cryo-electron microscopy (cryo-EM) [34] (Figure 2b). The transmembrane portion of Rcf2, including the HIG1 domain [36,51], forms extensive interactions with the transmembrane domains of Cox3 and Cox13, whereas the IMS α-helix, a unique feature of yeast HIGD proteins, interacts with Cox12 [34]. The relevance of the conserved HIG1 domain for Rcf2 protein function is highlighted by the observations that the Rcf2 HIG1 domain alone can interact with SCs in isolated mitochondria [61], and the expression of a truncated Rcf1 protein containing just the HIG1 domain is sufficient to support respiratory growth [36].

The Rcf2 arrangement described in vivo differs substantially from the structure of purified recombinant Rcf2 reconstituted in detergent micelles obtained by NMR [44,58]. Neither Rcf1 nor Rcf2 was soluble when expressed in *Escherichia coli* [58]. Once purified from inclusion bodies and refolded into detergent micelles, both proteins formed dimers, in which each monomer consists of five transmembrane α-helixes. Some of these helixes are unusually charged and form a zipper motive at the dimer interface [44,58]. In particular, the fifth transmembrane α-helix in the NMR structure corresponds to the C-terminal helix exposed to the IMS in the cryo-EM study (Figure 1c). Rcf dimerization could represent a way to shield the charged residues from the hydrophobic environment. In vivo, no Rcf1 dimers have been detected by crosslinking analysis [36]. However, we cannot exclude that Rcf proteins undergo conformational changes involving the flipping of the more hydrophilic C-terminal α-helix out of the inner membrane. Indeed the chemical shift changes detected in Rcf1 and Rcf2 upon the addition of lipids to the detergent micelles suggest that Rcf proteins can undergo structural rearrangements in response to the changing environment [44,58].

Efficient Rcf2 assembly into SCs requires Rcf1 [20]. Initially, Rcf1 and Rcf2 were not found to interact with each other physically, which suggested the existence of different SC populations [19]. However, both proteins and Rcf3 have been recently found to interact among themselves independently of SCs [61]. The decrease of SC-associated Rcf2 in the absence of Rcf1 could be the consequence of the Rcf1-dependent CIV assembly defect. Indeed, the incorporation of Cox13 into CIV occurs before the binding of Rcf2 [20], and a small fraction of Rcf2 was detected co-sedimenting with Cox12-Cox13-containing CIV species upon extraction with the mild detergent digitonin [36,61], but not n-Dodecyl β-D-Maltoside (DDM) [19], suggesting a transient or labile interaction. The stable presence of Rcf2 in hypoxic SCs suggests that, at least under certain conditions, this protein may be a stable SC component required to modulate cytochrome *c*-mediated electron transfer of CIV enzymatic activity in response to the cellular energetic requirements [34].

### 2.4. Role of Rcf Proteins under Hypoxia and Oxidative Stress

The MRC organization into SCs has a major physiological and biomedical relevance, as described elsewhere [15,62,63,64]. However, despite the proposal of multiple and controversial hypotheses, the proven functional benefits that the SC organization might provide to the cell are just starting to emerge. Electron flux through CIII and CIV can be limited by cytochrome *c* diffusion, and, therefore, minimizing the distance between these complexes is kinetically advantageous [57,65]. Moreover, the SC organization has been proposed to minimize ROS generation [66]. Given the roles of HIGD proteins in SC biogenesis and function, they could play relevant roles in adaptation to hypoxia or oxidative stress. Rcf1 and Rcf2 are constitutively expressed and do not respond to changes in oxygen concentration [51]. However, Rcf2 interacts with hypoxic SCs, as mentioned earlier.

Several observations have linked the Rcf proteins to a role in ROS control or adaptation to oxidative stress. The activity but not the protein levels of aconitase, an enzyme that contains Fe-S clusters sensitive to ROS, are significantly lower in Δrcf1 and Δ*rcf2* strains than in wild type or Δ*cox13* strains [20,50] (Cox13 is a non-essential CIV subunit). These data agree with the observation that H_2_DCFDA-reactive ROS levels are higher in the Δrcf1 and Δ*rcf2* strains than in the wild type and the Δ*cox13* and Δ*cox1* mutant strains [20]. Furthermore, the levels of mitochondrial superoxide dismutase (Sod2) are increased in the Δrcf1 strain, which is more sensitive than the wild type to exogenous H_2_O_2_ [50]. This evidence supports the role of Rcf1 and Rcf2 in optimizing electron transfer efficiency to minimize ROS generation. This phenotype would be, at least in part, dependent on their function in SC formation/activity because it is more pronounced in the Δ*rcf1* and Δ*rcf2* than in a Δ*cox13* strain, in which CIV activity is reduced but SC levels are not affected [20]. In addition, the respiratory growth defect of the Δ*rcf1* strain is exacerbated under hypoxia, whereas the respiratory growth of the Δ*cox13* strain is similar in the presence of 1% and 20% oxygen [20]. It is relevant in the context of stress that the Δ*rcf1* strain, but not the Δ*rcf2*, is thermosensitive [36,50], a phenotype that could be related to the inner membrane protein organization/disorganization when the levels of SCs are decreased.

Finally, two dimensional (2D) Blue Native/Sodium Dodecyl Sulfate (BN/SDS)-PAGE analyses of mitochondrial extracts prepared in the presence of digitonin have shown that the relative proportion of Rcf1 and Rcf2 found to co-sediment with SCs versus CIV-associated and free Rcf1/2 was significantly higher when cells were grown in respiratory rather than fermentable media, suggesting that Rcf proteins are recruited to SCs when a higher respiratory rate is required [36].

## 3. Role of HIGD1A and HIGD2A in Mammalian Cytochrome *c* Oxidase Biogenesis and Function

The mammalian HIGD-family proteins include two subgroups of yeast Rcf1 homologs, HIGD1A/B/C and HIGD2A/B. So far, only HIGD1A and HIGD2A have been identified as mitochondrial proteins and characterized in some detail.

HIGD1A and HIGD2A (of 10.4 and 11.5 kDa, respectively) have two hydrophilic transmembrane domains and are localized in the inner mitochondrial membrane [15,20,39,67,68], similarly to Rcf1. The structure of HIGD1A, determined by NMR [67], displayed an amphiphilic N-terminal helix presumably oriented parallel to the surface of the membrane (Figure 1d, top panel). Both transmembrane domains and the interhelical loop structures are very stable, probably because the latter is anchored into the membrane, whereas the C-terminal domain is unfolded and has significant fluctuations in its orientation [68]. The two HIGD1A transmembrane helices are loosely packed with interhelical distances of more than 8 Å and a few van der Waals contacts at the end of the helices, which is a common feature in proteins involved in signal transduction across membranes [67]. The structure of HIGD2A is yet to be elucidated, but it is predicted to be similar to HIGD1A (Figure 1d, middle panel). Although HIGD2A has an N-terminal domain more extended than HIGD1A, both proteins share conserved amino acids mainly located in the transmembrane and N-terminal regions [69] (Figure 1d, bottom panel), and their orientation exposes their N- and C-termini to the intermembrane space [15,20,39].

### 3.1. Regulation of HIGD Gene Expression

*HIGD1A* and *HIGD2A* transcripts have been detected in multiple mouse tissues and cell lines cultured in standard and a variety of stress conditions. In the mouse, *Higd1a* and *Higd2a* mRNAs were co-expressed mainly in the heart and kidney, but *Higd1a* mRNA was also abundant in the brain and *Higd2a* mRNA in leukocytes [69]. Furthermore, *Higd2a* expression has been related to cellular differentiation [29].

*HIGD1A* and *HIGD2A* expression is induced under stress conditions like hypoxia or glucose deprivation in human cervical epithelial cells and murine cerebral cortical neurons [30,69,70]. Higd1a also promotes the survival of murine pancreatic α and β cells under those stress conditions [39]. In fact, these proteins were initially identified in a screen for genes induced under hypoxia [71,72] and their protein levels increase in many cell types cultured at low oxygen concentrations [15,18,29,30,39,69,70], especially during the early stages of hypoxia [15,70], promoting cell survival [29,30,70]. The expression of *HIGD* genes depends on the binding of hypoxia-inducible transcription Factor HIF1α to their promoter regions [39,41,69], which contain potential hypoxic-responsive element (HRE) sites. In mice, Higd1a was also shown to be induced by HIF2α in macrophages [69], but not in embryonic fibroblasts (MEFs) or trophoblast stem cells [41]. Besides, the murine promoters of *Higd1a* and *Higd2a* were similarly responsive to CoCl2 and the iron chelator deferoxamine (DFO), which are able to mimic hypoxic conditions by inhibiting prolyl hydroxylase activity and stabilizing HIF. Remarkably, murine (but not human) *Higd2a* gene expression was repressed under hypoxic conditions in the presence of low glucose or galactose, indicating that its induction might be dependent on glucose availability [29].

The expression of *HIGD1A* and *HIGD2A* is not only regulated by HIF transcription factors. *HIGD2A* expression is also regulated by the binding of the transcription factor E2F1 to a (+)09A8446 site in the promoter region of the gene [29]. E2F1 has been related to several cellular processes, but it is primarily considered involved in regulating cell cycle progression, DNA-damage response, apoptosis, and global metabolism homeostasis [73]. E2F1 expression and action are linked to microRNAs through several feedback loops, in which microRNAs are activated by E2F1 and, simultaneously, the transcription factor is inhibited by different microRNAs. Remarkably, one of the microRNAs that directly targets and inhibits E2F1, called microRNA-372 [74], is predicted to target the positions 94–100 of the 3’UTR of *HIGD2A* (TargetScan [75]). *HIGD1A* mRNA levels can also be regulated by microRNAs. Specifically, microRNA-375 inhibits *HIGD1A* expression in pig Sertoli cells [76], inducing an increase in ROS production and apoptosis under low glucose conditions [76]. Other microRNAs are predicted to bind to 3’UTR regions of *HIGD1A* and *HIGD2A*, which increases the complexity of regulation of these genes. However, further studies are necessary to validate those predictions and establish a transcriptional regulatory model for *HIGD* genes, which could provide further information about their functions.

Whether additional stressors influence the expression of *HIGD* genes has not yet been fully explored. However, it is known that *HIGD1A* and *HIGD2A* expression are not induced by the presence of hydrogen peroxide [69], except under high-fat exposure [77]. Finally, it has been reported that murine *Higd1a* is upregulated in the presence of nitric oxide (NO) [69].

### 3.2. Role of HIGD Proteins in the Assembly and Regulation of Mammalian Cytochrome c Oxidase

HIGD1A and HIGD2A were initially described in mammalian cellular models as hypoglycemia/hypoxia-inducible mitochondrial proteins that promote cell survival under these stresses [39,69]. Ectopic expression of Higd1a in mouse macrophages was reported to decrease apoptosis, inhibiting the release of cytochrome *c* from the mitochondria and reducing caspase activity [69]. Its function was also linked to mitochondrial fission and organization of the cristae through its interaction with the Optic Atrophy Type 1 (OPA1) mitochondrial dynamin-like GTPase [78], and the maintenance of the mitochondrial function by inhibiting the mitochondrial gamma-secretase [79]. Additionally, both HIGD proteins have been shown to interact with different MRC complexes [15,16,17,31,70]. Here, we will focus exclusively on the role of HIGD1A and HIGD2A in the regulation of the biogenesis and activity of cytochrome *c* oxidase.

#### 3.2.1. Role in the Assembly and Regulation of Cytochrome *c* Oxidase in Physiological Conditions

Following the discovery of Rcf1 and Rcf2 in yeast, Chen and colleagues showed that silencing Higd2a in mouse myoblasts resulted in the attenuation of CIV-containing supercomplex levels, as seen in the Δrcf1 yeast strain [50]. In contrast, *Higd1a* knockdown did not affect the levels of CIV or respiratory SCs. Analyses in living human HeLa cells using fluorescence lifetime imaging microscopy (FLIM) and Förster Resonance Energy Transfer (FRET) analyses showed that *HIGD2A* silencing resulted in the release of CIV from the SCs [80]. These results suggested functional conservation between Rcf1 and HIGD2A, which was confirmed by the ability of *HIGD2A* to suppress the respiratory growth defect observed in Δ*rcf1* and Δ*rcf1*Δ*rcf2* yeast mutant strains [15,20]. Until recently, this extrapolation of HIGD2A function from yeast studies was the only information about the role of HIGD2A. However, thanks to the development of gene-editing technologies, two independent studies have reported the generation of knockout (KO) cell lines for *HIGD1A* and *HIGD2A* [15,18], whose characterization allowed to gain insight into the roles of these proteins. Similar to yeast Rcf1, the primary function of HIGD2A is to assist CIV biogenesis. The two reports have shown that the absence of *HIGD2A* produces a decrease in the levels of CIV and CIV-containing SCs, together with an accumulation of CIV assembly intermediates [15,18]. The CIV assembly deficiency of *HIGD2A*-KO cells leads to defective cytochrome *c* oxidase enzymatic activity and, therefore, impaired cell respiration rate.

The mechanisms of action of yeast Rcf1 and mammalian HIGD2A are very alike. First, HIGD2A also acts as a CIV assembly factor by interacting with newly synthesized COX3 subunit to chaperone its modular assembly and incorporation into the holoenzyme [15,18] (Figure 3a). In the absence of HIGD2A, the COX3-assembly module cannot form or becomes unstable, which results in decreased steady-state levels of its component subunits (COX3, COX6A1, COX76B1, COX7A2, and NDUFA4) [18], and the accumulation of the COX1- and COX1+COX2-assembly modules [15]. Second, the conserved QRRQ motif of HIGD2A is necessary for its role in CIV assembly and maturation, and is required for its binding to newly-synthesized COX3 and holo-CIV [15]. Third, HIGD2A forms a complex with three nucleus-encoded subunits (COX4-1, COX5B, and COX6A1), relatively comparable to the complex that Rcf1 forms with Cox7 (human COX7A), Cox4 and Cox13 (the yeast homologs of COX5B and COX6A1), and Cox3 [21]. However, these complexes differ in that the three yeast subunits binding to Rcf1 are COX3-module subunits, whereas the human HIGD2A interactors COX4-1, COX5B, and COX6A1 are subunit components of the COX1, COX2, and COX3 assembly modules, respectively.

This HIGD2A complex, of 50 kDa, has been proposed to coordinate the assembly of the three modules by the release of HIGD2A when necessary to assemble and incorporate the COX3 module [15] (Figure 3a). Adding complexity to the CIV assembly pathway, the human subunit COX5B can assemble with HIGD2A and the COX3 module or with the COX2 module (Figure 3a), whereas the yeast homolog of COX5B, Cox4, is a constituent of both the Cox3 and Cox1 modules. This Cox4/COX5B promiscuity has suggested that this subunit could be incorporated into CIV by two alternative pathways [15,21,82], emphasizing that multiple pathways may have evolved to facilitate CIV biogenesis. Fourth, whereas in the absence of CIV, yeast Rcf1 interacts with CIII [19,20,50,61], HIGD2A has been found to interact with subunits of CI and CIII [15,18]. This has suggested a role for HIGD2A in the incorporation of the COX3 module not only to isolated CIV but also directly into the SCs (Figure 3b). In fact, the COX3 subunit was absent in the SC “I–III_2_–IV”-like present in *HIGD2A*-KO cells [18]. Moreover, by complexome analysis and import of in vitro synthesized [[35] S]-HIGD2A into isolated energized mitochondria followed by BN-PAGE analyses, it was also observed that the 50-kDa HIGD2A regulatory complex, together with other CIV subunits, could directly incorporate into the SC I–III_2_ in COX1 and COX2 cybrid cell lines [15,16]. Whereas in the COX1 cybrid line, the incorporation of the 50-kDa HIGD2A complex into SC I–III_2_ becomes rapidly unstable, it forms a more stable intermediate in the absence of COX2, that was referred to as SC I–III_2plus_ [16]. Therefore, COX1 seems to be essential for the biogenesis of CIV in the context of SCs. Although radiolabeled recombinant HIGD2A is found associated with SCs upon import into wild type mitochondria, and significant amounts of the endogenous protein associate to SCs in cybrid CIV mutant cell lines, only minute steady-state levels of HIGD2A are found bound to SCs in mitochondria purified from wild type HEK293T or 143B human cells. This suggests that either HIGD2A is released once CIV and SCs are fully assembled, or the interaction is very labile. Either possibility could explain why HIGD2A has not been found in the mammalian SC structures. In summary, similar to yeast Rcf1, HIGD2A functions to promote COX3-module formation and regulate the overall CIV modular assembly, both as a free or supercomplexed holoenzyme (Figure 3).

HIGD1A is evolutionary more distant from Rcf1 (Figure 1), and when heterologously expressed in Δrcf1 or Δrcf1Δrcf2 yeast strains, it was not able to substitute for the function of Rcf proteins [15,50]. By complexome analysis of human mitochondrial extracts separated by BN-PAGE, HIGD1A was shown to form a cluster with two early assembly CIV subunits, COX4-1 and COX5A, [17], an interaction that has been confirmed by immunoprecipitation assays in several studies [15,17,70]. Moreover, some COX1 module subunits have been found decreased in *HIGD1A*-KO cells [15,18], both by immunoblotting and by mass spectrometry, probably because of the destabilization of the early assembly cluster that it forms. However, the absence of HIGD1A protein leads to no changes in CIV levels, assembly kinetics, or its organization into supercomplexes [15,18]. *HIGD1A*-KO cells do not accumulate CIV intermediates, and the primary function of the HIGD1A protein in respiratory complex assembly is related to CIII maturation [15]. Nevertheless, the absence of HIGD1A decreases cytochrome *c* oxidase activity, even though its effect is not as dramatic as in the absence of HIGD2A [15,70]. In agreement, chimeric maltose-binding protein(MBP)-tagged HIGD1A was shown to interact with cytochrome *c* by isothermal titration calorimetry and immunoprecipitation assays and to increase bovine CIV enzymatic activity in vitro [31]. These results have suggested the possibility of a HIGD1A-mediated electron transfer bridge between CIII and CIV in the context of SCs, comparable to the model proposed for Rcf1 (Figure 4).

Remarkably, the N-terminal domain of HIGD1A, exposed to the IMS, was not necessary for its interaction with cytochrome *c*. Similarly, recombinant rat Higd1a was shown to interact with purified bovine CIV enzyme in vitro, producing an alteration in the hemes absorption spectra that indicated a possible conformational change affecting the catalytic core of the enzyme [70]. Using resonance Raman spectroscopy, the authors showed that the addition of Higd1a produces a partial conversion of the heme *a* in COX1 from a low-spin to a high-spin state, which indicates a structural modification of the active center of the enzyme, increasing CIV activity [70]. A simulation via COOT software [83] helped the authors to predict the interaction of the Higd1a TM1 with the COX1 TM11 and TM12, affecting the structure of COX1 Arg38 and the formyl group of heme *a* and accelerating the proton-pumping H pathway. Currently, there is no structural data to support these findings. However, the same research group corroborated both the interaction of Higd1a with CIV and the regulation of the activity of the enzyme depending on the levels of Higd1a in vivo using rat cardiomyocyte [70].

These data discard a prominent role of HIGD1A as a CIV assembly factor since its loss does not produce significant changes in CIV composition or distribution as a free or supercomplexed structure. Rather, the current data suggest a function for HIGD1A as a modulator or regulator of cytochrome *c* oxidase activity by two mechanisms: (1) the binding to cytochrome *c* to facilitate its interaction with CIV, acting as an electron transfer bridge; and (2) the structural alteration of the active center heme *a* of CIV to increase the proton-pumping and, then, ATP production. In contrast, in the *HIGD2A*-KO cell line, there is a good correlation between CIV steady-state levels and activity; thus, supporting a role for HIGD2A in regulating CIV assembly, but not in modulating the catalytic function of assembled CIV.

#### 3.2.2. Role in the Assembly and Regulation of Cytochrome *c* Oxidase under Hypoxic Conditions

Investigating the role of HIGD proteins in the regulation of cytochrome *c* oxidase activity under hypoxia is of high biological significance because: (1) HIGD1A and HIGD2A belong to the Hypoxia Inducible Gene Domain family and their expression is activated under low oxygen levels; and (2) Cytochrome *c* oxidase is the major oxygen consumer in the cell, and it has evolved to sense and adapt to environmental oxygen levels. Therefore, HIGD proteins might play a role in the adaptation of this enzyme to hypoxic conditions.

The absence of *HIGD1A* or *HIGD2A* destabilizes the mitochondrial respiratory complexes under acute or chronic hypoxia. However, a role for HIGD proteins on the assembly of SCs, specifically in these conditions, has been discarded by one group [18]. On the contrary, another found that, under low oxygen levels, a condition in which *HIGD1A* expression is induced, the CIV assembly defect of *HIGD2A*-KO cells was attenuated. This restoration of CIV assembly was also observed when HIGD1A was overexpressed in cells lacking HIGD2A, conditions in which HIGD1A was detected in a ~50 kDa intermediate and was able to interact with COX6A1. These results could indicate either the recruitment of COX6A1 to the HIGD1A-COX4-1-COX5A early assembly cluster, or the recruitment of HIGD1A to the COX4-1-COX5B-COX6A1 module. Either way, HIGD1A may play a possible overlapping role with HIGD2A in CIV assembly in conditions in which it is overexpressed, such as hypoxia or oxidative stress [15]. In addition to its possible role in the assembly of CIV, HIGD1A was also shown to regulate CIV activity during hypoxia. Using a FRET-based approach to measure ATP concentration in rat cardiomyocytes, it has been shown that decreasing Higd1a levels result in lowered ATP production and compromised cell viability, whereas the overexpression of Higd1a increased the levels of ATP while reducing cell death, without affecting SC levels [70].

These data suggest a role of HIGD1A in CIV regulation and assembly that might be redundant or not necessary in physiological conditions but essential for the adaptation and/or restoration of CIV activity under certain stressor environmental conditions (e.g., mutations or hypoxia). The fact that HIGD1A interacts with the CIV subunit COX4-1 suggests other possibilities for the role of HIGD1A in CIV regulation. Mammalian COX4 subunit exists in two tissue- and condition-specific isoforms: a normoxic COX4-1 isoform expressed in most tissues and a hypoxic COX4-2 isoform expressed in the lungs and under low oxygen levels that enhances CIV activity [84]. Thus, HIGD1A might also be implicated in the CIV subunit composition switch from COX4-1 to COX4-2, a possibility that remains to be examined.

## 4. Concluding Remarks

Members of the HIGD family of proteins are found in bacteria through eukaryotes. They have been shown in diverse organisms to be upregulated during hypoxic, hypoglycemic, and oxidative stress conditions, in which they promote cell survival. Among the eukaryotic members of the family, several have been found to localize to mitochondria where they regulate the biogenesis, organization, and function of the respiratory chain enzymatic complexes. Yeast Rcf1/2 and mammalian HIGD1A/2A play distinct and multifarious roles in these processes, several of which directly affect cytochrome *c* oxidase or respiratory chain complex IV.

Yeast Rcf1 and human HIGD2A act as classical CIV assembly factors and function to promote the formation and assembly of the COX3 module. They are not essential in standard physiological conditions, as the respective KO strains and cell lines are able to assemble some functional CIV, most probably sustained by the functional recruitment of their family isoforms. However, their presence not only fuels CIV biogenesis efficiently, but may also allow for the formation of heterogeneous CIV species or the assembly of the complex directly in the context of respiratory supercomplexes. This may have important implications on how the cell adapts to changing metabolic or environmental conditions, which are yet to be explored. It also has implications in the mechanism of SC assembly since it suggests that SCs do not necessarily form by the coming together of preassembled individual complexes. What exactly triggers supercomplexed versus free CIV assembly remains to be understood.

Yeast Rcf2/3 and human HIGD1A may also participate in CIV biogenesis, particularly under hypoxia, oxidative or metabolic stress. However, together with Rcf1, they play fundamental roles in the modulation of CIV enzymatic activity without altering the supercomplex organization. In yeast, overexpression of Rcf1 or the absence of Rcf2 or Rcf3 increases oxygen flux through the respiratory chain by upregulation of the CIV activity [61]. Similarly, overexpression of HIGD1A enhances CIV activity. In both organisms, the mechanisms involve interactions of the HIGD proteins with CIV. In yeast, Rcf1 additionally interacts with cytochrome *c* and CIII, and it has been proposed to establish a bridge that facilitates electron transfer. This model needs to be substantiated in whole cells but is supported by the recent observation that yeast SCs III_2_+IV_1-2_ provide a bioenergetics advantage by minimizing the distance of cytochrome *c* diffusion between CIII and CIV [57].

Further investigations and structural data are necessary to elucidate the mechanism of action of human HIGD proteins in CIV regulation under low oxygen levels and other stress conditions that activate their expression. This could have important implications for human diseases. For example, hypoxia regimens effectively improved survival and attenuated symptoms in animal models of mitochondrial diseases [85,86]. Similarly, the expression of HIGD1A in different cellular and animal models of mitochondrial disease was shown to attenuate their deleterious phenotypes by increasing CIV activity and ATP production while decreasing ROS levels [32]. Consequently, elucidating the mechanism of action of HIGD proteins under low oxygen levels could reveal new potential therapeutic candidates, mimicking or enhancing the HIGD1A mechanism of action.

Human HIGD1B/C and HIGD2B remain uncharacterized. Nothing is known regarding their tissue distribution, intracellular location, and function and whether they are constitutively expressed or respond to specific stimuli. Their study is imperative to achieve a full understanding of the independent and overlapping functions performed by human HIGD proteins.

Finally, HIGD proteins have been shown to interact with proteins other than respiratory chain complex subunits, such as the ADP/ATP carrier, and with lipids such as cardiolipin [19,33]. The possibility that, through these interactions, the HIGD proteins coordinate CIV and whole respiratory chain assembly and function with general mitochondrial membrane biogenesis and global cellular metabolism deserve future investigations.

## Figures and Tables

**Figure 1 cells-09-02620-f001:**
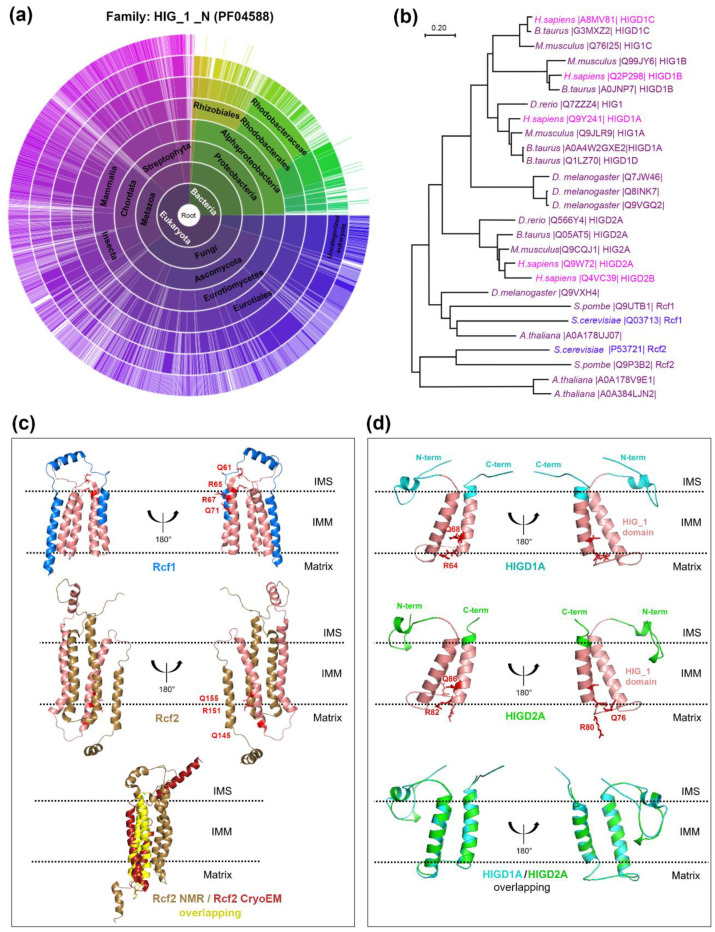
Distribution of *HIG_1_N* (PF04588) domain proteins across species. (**a**) Modified “sunburst” visualization of the taxonomic lineage distribution of 1911 different species that in total have 3472 protein sequences containing the hypoxia-inducible gene (HIG)-1-N domain (Pfam ID: PF04588). The graph shows each node in the tree as a separate arc, arranged radially with the superkingdoms at the center and the species arrayed around the outermost ring. The graph was generated with tools from pfam.xfam.org hosted by the European Bioinformatics Institute at the European Molecular Biology Laboratory (EMBL-EBI). Yellow-green colors represent different types of bacteria, and purple color represent eukaryotes. (**b**) Phylogenetic tree of Hypoxia-inducible gene domain (HIGD) proteins across eukaryotic species. The evolutionary history was inferred using the Neighbor-Joining method. We show the optimal tree with the sum of branch length = 8.97673253. The tree was drawn to scale, with branch lengths in the same units as those of the evolutionary distances used to infer the phylogenetic tree. The evolutionary distances were computed using the Poisson correction method and are in the units of the number of amino acid substitutions per site. This analysis involved 27 amino acid sequences obtained from UniProt and entered manually. All ambiguous positions were removed for each sequence pair (pairwise deletion option). There were a total of 253 positions in the final dataset. Evolutionary analyses were conducted in MEGA X [42,43]. (**c**) Solution NMR structures of *S. cerevisiae* respiratory supercomplex factors Rcf1 and Rcf2. The top panel shows ribbon diagrams of the Rcf1 structure (Protein database -PDB- code 5NF8), with the HIG_1 domain in salmon. The middle panel shows the Rcf2 structure (PDB 6LUL) [44], with the HIG_1 domain in salmon. The QRRQ motifs in Rcf1 and Rcf2 are marked in red. The lower panel shows the overlapping of the NMR structure of Rcf2 (PDB 6LUL) (in gold) and the partial cryo-EM structure for Rcf2 as bound to hypoxic respiratory supercomplex (SC) CIII_2_+CIV (PDB 6T15) [34] (in red). The Rcf2 transmembrane helices resolved by NMR that overlap with the Rcf2 cryo-electron microscopy (cryo-EM) structure are presented in yellow. A prominent difference between the two structures relates to the C-terminal α-helix, shown to protrude into the intermembrane space by cryo-EM. (**d**) Solution NMR structures of human HIGD1A and HIGD2A. The top panel shows ribbon diagrams of the structure of HIGD1A (PDB 2LOM), with the HIG_1 domain in salmon. HIGD1A is a type 1 HIGD protein, with a truncated QRRQ motif, labeled in red. The middle panel presents ribbon diagrams of the predicted structure of HIGD2A. The structure was predicted in silico using Swiss-Model based on the alignment with HIGD1A, with 0.75 coverage of the sequence [45,46,47]. Residues of the QRRQ motif are labeled in red and HIG_1 domain, in salmon. The bottom panel depicts a superposition view of HIGD1A and HIGD2A structures generated by PyMOL. IMS: intermembrane space. IMM: inner mitochondrial membrane.

**Figure 2 cells-09-02620-f002:**
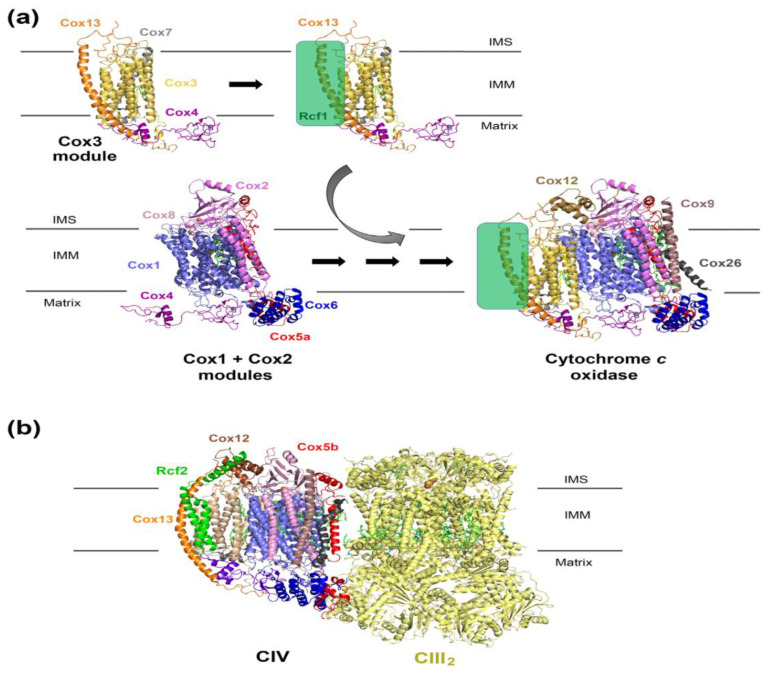
Proposed role of Rcf1 in the regulation of *S. cerevisiae* cytochrome *c* oxidase assembly and activity. (**a**) Schematic view of the yeast *S. cerevisiae* modular cytochrome *c* oxidase assembly pathway [21], depicting the role of Rcf1. Rcf1 interacts with the Cox3 and Cox13 in the Cox3-assembly module and promotes its incorporation into the CIV assembly line to join the assembled Cox1 + Cox2 module. The presence of Rcf1 facilitates the incorporation of Cox12. The figure was generated with PyMOL using the *S. cerevisiae* cytochrome *c* oxidase cryo-EM structure PDB 6YMY [57]. Rcf1 was incorporated as a green box, interacting with Cox3 and Cox13 as shown by biochemical studies. (**b**) Structure of the *S. cerevisiae* hypoxic respiratory SC III_2_–IV containing the Cox5 hypoxic isoform Cox5b and Rcf2 (PDB 6T15) [34].

**Figure 3 cells-09-02620-f003:**
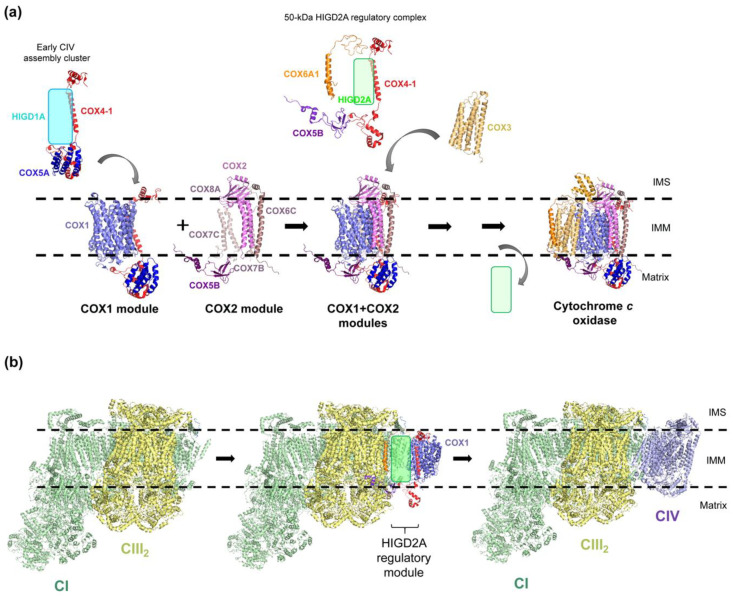
Roles of HIGD1A and HIGD2A in cytochrome *c* oxidase assembly. (**a**) Schematic view of the mammalian modular cytochrome *c* oxidase assembly pathway. HIGD1A is present in an early assembly cluster, together with the COX1 module proteins COX4-1 and COX5A. HIGD2A forms a 50 kDa regulatory module with COX4-1, COX5B, and COX6A1, which binds to newly synthesized COX3 to promote its binding to the previously assembled COX1 + COX2 module. HIGD1A and HIGD2A are represented by a blue and a green box, respectively, due to the lack of structural information. (**b**) The HIGD2A regulatory module plays a role in supercomplexed cytochrome *c* oxidase assembly, by interacting with SC I+III_2_ and facilitating the incorporation of the remaining CIV subunits. The figure was generated with PyMOL and Adobe Photoshop using the structure of the CI-CIII_2_-CIV respirasome from ovine heart mitochondria [81] (PDB 5J4Z). IMS: intermembrane space. IMM: inner mitochondrial membrane. CI: complex I. CIII_2_: complex III dimer. CIV: complex IV.

**Figure 4 cells-09-02620-f004:**
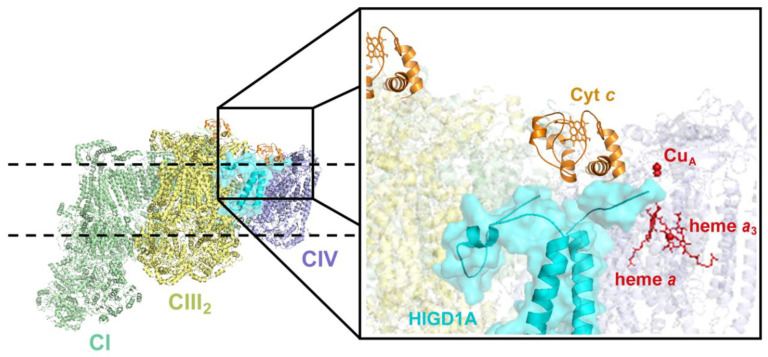
HIGD1A interacts with CIV and enhances its activity. Following a proposed model [31,70], overexpressed HIGD1A interacts with cytochrome *c* and CIV and produces a structural change in the CIV heme *a* active center, acting as a positive modulator of the enzyme. The figure was generated with PyMOL and Adobe Photoshop using the structure of the CI-CIII_2_-CIV respirasome from ovine (*Ovis aries*) heart mitochondria [81](PDB 5J4Z). IMS, intermembrane space. IMM, inner mitochondrial membrane. CI, complex I. CIII_2_, complex III dimer. CIV, complex IV.

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
