# Peer review of "HIGD-Driven Regulation of Cytochrome c Oxidase Biogenesis and Function"

_cells, 2020, doi:10.3390/cells9122620_

Round 1

Reviewer 1 Report

The manuscript is descriptive, well written and represents an overview of the mechanisms underlying the CIV biogenesis control and the role of HIGD proteins in regulation of the mitochondrial respiratory chain complex IV functioning. The role of Rcf proteins was discussed and in what positive/negative manners the absence/presence of these proteins  modulate the assemby, interactions and functioning of CIV. Several lines of evidence were analyzed to draw the conclusion on the association of Rcf proteins with the mitochondrial chain biogenesis. In addition the mammalian proteins HIGD1A and HIGD2A are also insepcted for their role similar to Rcf proteins. The regulations of HIGD1A and HIGD2A expression is commented and the factors influencing the expression. The two HIGD proteins were subjected to analyses of their influence on the citochrome c oxidase and in what manner they interact. Distinct differences were recognized between the yeast RCf1 and 2 and the mammalian higd1a and 2 proteins  in their influence on the mitochondrial CIV chain complex and not all are fundamental in the modulation of CIV enzymatic activity. Manuscript is of value to point out the most important references on the case of the factors and mechanisms shaping the CIV biogenesis, interactions, assemblies and functioning. For this reason, it is acceptable for publishing in its form representing a stand point on the matter by the gorup of coauthors.

Author Response

We appreciate the kind comments from the reviewer.

Reviewer 2 Report

The manuscript, entitled “HIGD-driven regulation of cytochrome c oxidase 2 biogenesis and function”, was focused on the regulated roles of the Hypoxia Inducible Gene Domain (HIGD) family either yeast  respiratory supercomplex factors 1 and 2 (Rcf1 and Rcf2) or two mammalian homologs of Rcf1 (the proteins HIGD1A and HIGD2A) in physiological and stress conditions such as cytochrome c oxidase assembly and supercomplex biogenesis, hypoxia and oxidative stress in yeast and  mammalian mitochondria by reviewing and analyzing the structures of cytochrome c oxidase assembly as well as their activities. The cited materials are detailed and accurate. The layout arrangement is scientific, reasonable and easy to understand. I am happy to learn lots form the review. My only suggestion is to pay attention to the format of special indentation for each paragraph.

Author Response

The paragraph format has been revised for accuracy.

Reviewer 3 Report

This is an very detailed and extensive review of  the role of HIGD- in cytochrome c oxidase regulation biogenesis and function. It provides ample information, detailed figures and is very well written.

I have only two minor questions;

1-re cytochrome c oxidase activity; is the activity / are kinetics different  in monomers / dimers/supercpomplexes and in the presence or absence of HIGD/Rcf?

2- is there any human genetic disease conditions where HIGD 1/2/ are invoved ?

Author Response

1-re cytochrome c oxidase activity; is the activity / are kinetics different in monomers/ dimers/supercomplexes and in the presence or absence of HIGD/Rcf?

In wild-type yeast mitochondria, all the CIV present is associated with CIII into SCs, whereas free monomeric CIV accumulates only when SC assembly is affected, as for CIII-deficient mutants. Dimeric CIV is not present in yeast. We and others have reported that CIV enzymatic activity is not severely compromised in strains lacking CIII. It is neither compromised in strains in which its association into SC is prevented and monomeric CIV accumulates at steady-state levels comparable to wild-type. Otherwise, CIV enzymatic properties are affected by the absence of Rcf1, as lengthily discussed in section #2.2 of the manuscript. However, it is difficult to determine if Rcf1 plays a direct role in modulating CIV activity or the observed functional alterations are rather a consequence of a CIV assembly defect.

In mammalian cells, Shinzawa-Itoh et al (PNAS, 2019) attempted to distinguish potential differences in cytochrome coxidase (CIV) activity in its different organizations. The authors showed that the monomeric form of bovine CIV was the activated form, compared to its dimeric form. We have included this information in the introduction of the revised manuscript. However, there are no published results regarding the difference in activity of the different CIV forms in the absence or presence of HIGD proteins. Since HIGD1A is not present in either CIV or CIV2, its effect could only be assessed in supercomplexes. In the case of HIGD2A, it will be difficult to differentiate its role in assembly and activity because, as explained in the text: “in the HIGD2A-KO cell line, there is a good correlation between CIV steady-state levels and activity, thus supporting a role for HIGD2A in regulating CIV assembly but not in modulating the catalytic function of assembled CIV”.

2- is there any human genetic disease conditions where HIGD 1/2/ are involved?

To present, no patient has been described with mutations/abnormalities in HIGD1A or HIGD2A.

HIGD1A promotes has been involved in inflammatory and hypoxia-related diseases, including atherosclerosis, ischemic heart disease, and Alzheimer’s disease as well as cancer. However, in most cases, the severity of the disease correlates with a localization of the protein to the nucleus and cell death, but there is no evidence that HIGD1A role on CIV assembly or activity, the focus of this review, is involved in any way. Therefore, we have not included any comment on this.